# DISCRIMINABILITY DISTILLATION
# IN GROUP REPRESENTATION LEARNING

## ABSTRACT

Learning group representation is a commonly concerned issue in tasks where the basic unit is a group, set or sequence. Previously, the computer vision community tries to tackle it by aggregating the elements in a group based on an indicator either defined by human such as the quality or saliency of an element, or generated by a black box such as the attention score or output of a RNN.

This article provides a more essential and explicable view. We claim the most significant indicator to show whether the group representation can be benefited from an element is not the quality, or an inexplicable score, but the *discrimiability*. Our key insight is to explicitly design the *discrimiability* using embedded class centroids on a proxy set, and show the discrimiability distribution *w.r.t.* the element space can be distilled by a light-weight auxiliary distillation network (DDNet). This procedure is denoted as *discriminability distillation learning* (DDL). We show the proposed DDL can be flexibly plugged into many group based recognition tasks without influencing the training procedure of the original tasks. Comprehensive experiments on set-to-set face recognition and action recognition valid the advantage of DDL on both accuracy and efficiency, and it pushes forward the state-of-the-art results on these tasks by an impressive margin.

## 1 INTRODUCTION

With the rapid development of deep learning and the easy access to large-scale group data, recognition tasks using group information have drawn great attention in the computer vision community. The rich information provided by different elements can complement each other to boost the performance of tasks such as face recognition, action recognition, and person re-identification (Wang et al., 2017b; Zhong et al., 2018; Girdhar et al., 2017; Simonyan & Zisserman, 2014; Yang et al., 2017; Liu et al., 2019a; Rao et al., 2017b). While traditional practice for group-based recognition is to either aggregate the whole set by average (Li et al., 2014; Taigman et al., 2014) or max pooling (Chowdhury et al., 2016), or just sampling randomly (Wang et al., 2016), the fact that certain elements contribute negatively in recognition tasks has been ignored. Thus, an important issue is to select representatives from sets for efficient group understanding.

To tackle such cases, previous methods aim at defining the "quality" or "saliency" for each element in a group (Liu et al., 2017c; Yang et al., 2017; Rao et al., 2017b; Nikitin et al., 2017). The weights for each element can be automatically learned by self-attention. For example, Liu et al. (2017c) proposes the Quality Aware Network (QAN) to learn quality score for each image inside an image set during network training. Other works adopt the same idea and extend to specific tasks such as video-based person re-identification (Li et al., 2018; Wu et al., 2018) and action recognition (Wang et al., 2018c) by learning spatial-temporal attentions. However, the whole online quality or attention learning procedures are either manually designed or learned through a black box, which lacks explainability.

In this work, we explore deeper into the underlying mechanism for defining effective elements instead of relying on self-learned attention. Assuming that a base network has already been trained for element-based recognition using class labels, we define the "discriminability" of one sample by how difficult it is for the network to discriminate its class. As pointed out by Liu et al. (2018) that the feature embedding of elements lies close to the centroid of their corresponding class are the representatives, while features far away or closer to other classes are the confusing ones which

are not discriminative enough. Inspired by this observation, we identify a successful discriminability indicator by *measuring one embedding's distance with class centroids and compute the ratio of between positive and hardest-negative,* where the positive is its distance with its class's corresponding centroid and the hardest-negative is the closest counterpart. This indicator is defined as the discriminability distillation regulation (DDR).

Armed with recent theories on the homogeneity between class centroids and projection weights of classifiers (Wang et al., 2017a; Liu et al., 2017b; 2018; Deng et al., 2019a), the entire distance-measuring procedure can be easily accomplished by simply encoding all elements in one group. Thus, the DDR scores can be assessed for each element after the training of the base network. This assessing procedure is highly flexible without human supervision nor re-training the base network, so it can be adapted to any existing base. With our explicitly designed discriminability indicator on the training set, the distillation of such discriminability can be successfully performed with a light-weight discriminability distillation network (DDNet), which shows the superiority of our proposed indicator. We call the whole procedure uniformly as *discriminability distillation learning* (DDL).

The next step is towards finding a better aggregation policy. At the test phase, all elements are firstly sent to the light-weight DDNet. Then element features will be weighted aggregated by their DDR score into group representation. Moreover, in order to achieve the trade-off between accuracy and efficiency, we can filter elements by DDR score and only extract element features of high score. Since the base model tends to be heavy, the filter can save much computation consumption. We evaluate the effectiveness of our proposed DDL on several classical yet challenging tasks. Comprehensive experiments show the advantage of our method on both recognition accuracy and computation efficiency. We achieve state-of-the-art results without modifying the base networks.

We highlight our contributions as follows: (1) We define the *discriminability* of one element within a group from a more essential and explicable view, and propose an efficient indicator. (2) We verify that a light-weight network has the capacity of distilling discriminability from the assessed elements. Combining the post-processing with the network, the great computation burden can be saved comparing with existing methods. (3) We validate the effectiveness of DDL for both efficiency and accuracy on set-to-set face recognition and action recognition through extensive studies. State-of-the-art results can be achieved.

## 2 RELATED WORK

### 2.1 SET-TO-SET RECOGNITION

Set-to-set recognition which utilizes a group of data of the same class, has been proved efficient on various tasks and drawn much attention these years since the more videos and group datasets are available. Compared with recognition with a single image, set-to-set recognition can further explore the complementary information among set elements and benefit from it. Particularly in this paper, we care for the basic task of face and action recognition.

**Face Recognition.** To tackle set-to-set face recognition problem. (Wolf et al., 2011; Kalka et al., 2018; Beveridge et al., 2013; Klare et al., 2015), traditional methods directly estimate the feature similarity among sets of feature vectors (Arandjelovic et al., 2005; Harandi et al., 2011; Cevikalp & Triggs, 2010). Other works seek to aggregate element features by simply applying max pooling (Chowdhury et al., 2016) or average pooling (Li et al., 2014; Taigman et al., 2014) among set features to form a compact representation. However, since most set images are under unconstrained scenes, huge variations on blur, resolution, and occlusion appear, which will degrade the set feature discrimination. How to design a proper aggregation method for set face representation has been the key problem for this approach.

Recently, a few methods explore the quality or attention mechanism to form set representation. GhostVLAD (Zhong et al., 2018) improves traditional VLAD and down weight low quality element features. While Rao et al. (2017a) combine LSTM and reinforcement learning to discard low quality element features. Liu et al. (2017c) and Yang et al. (2017) introduce attention mechanisms to assign quality scores for different elements and aggregate feature vectors by quality weighted sum. To predict the quality score, an online attention network module is added and co-optimized by the target set-to-set recognition task. However, the definition of generated 'quality' score remains unclear and

the learning procedures are learned through a black box, which lacks explainability. In our work, we claim that the most significant indicator to show whether the group representation can be benefited from an element is not the quality or an inexplicable score, but the discriminability. And a novel discriminability distillation learning procedure is proposed.

**Action recognition.** With the advance of multimedia era, millions of hours of videos are uploaded to video platforms every day, so video understanding task like action recognition has become a popular research topic. Real-world videos contain variable frames, so it is not practical to put the whole video to a memory limited GPU. The most usual approach for video understanding is to sample frames or clips and design late fusion strategy to form video-level prediction.

Frame-based methods (Yue-Hei Ng et al., 2015; Simonyan & Zisserman, 2014; Girdhar et al., 2017) firstly extract frame features and aggregate them. Simonyan & Zisserman (2014) propose the two-stream network to simultaneously capture the appearance and motion information. Wang et al. (2017b) add attention module and learn to discard unrelated frames. Frames-based methods are computational efficient, but only aggregate high-level frame feature tends to limit the model ability to handle complex motion and temporal relation.

Clip-based method (Tran et al., 2015; 2018; Feichtenhofer et al., 2018) use 3D convolutional neural network to jointly capture spatio-temporal features, which perform better on action recognition. However, clip-based methods highly rely on the dense sample strategy, which introduces huge computational consumption and makes it unpractical to application. In this article, we show that by combing our DDL, the clip-based methods can achieve both excellent performance and computation efficiency.

# 3 DISCRIMINABILITY DISTILLATION LEARNING

In this section, we first formulate the problem of group representation learning in section 3.1 and then define the discriminability distillation regulation (DDR) in section 3.2. Next, we introduce the whole discriminability distillation learning (DDL) procedure in section 3.3. In sections 3.4 and 3.5, we discuss the aggregation method and the advantage of our DDL, respectively.

## 3.1 FORMULATION OF GROUP REPRESENTATION LEARNING

Group representation learning focuses on formulating a uniform representation for a whole set of elements. The core of either verification task or classification task is how to aggregate features of a given element group.

Define $f_i$ as the embedded feature of element $I_i$ in a group $I_S$, then the uniform feature representation of the whole group is

$$F_{I_S} = \mathcal{G}(f_1, f_2, \cdots, f_i), \tag{1}$$

where $\mathcal{G}$ indicates the feature aggregation module. While previous research has revealed that conducting $\mathcal{G}$ with quality scores (Liu et al., 2017c) has priority over simple aggregation, this kind of methods is not explainable. In this article, we propose a discriminability distillation learning (DDL) process to generate the *discriminability* of feature representation.

## 3.2 DISCRIMINABILITY DISTILLATION REGULATION

Towards learning efficient and accurate $\mathcal{G}$, we design the discriminability distillation regulation (DDR) to generate the *discriminability* to replace the traditional quality score. In DDR, we jointly consider the feature space distribution and explicitly distill the *discriminability* by encoding the intra-class distance and inter-class distance with class centroids. Let $\mathcal{X}$ denote the training set with the identities $C$ and $W_j, j \in [1, C]$ is the class centroid. For feature $f_i, i \in [1, s]$ with class $c$ where

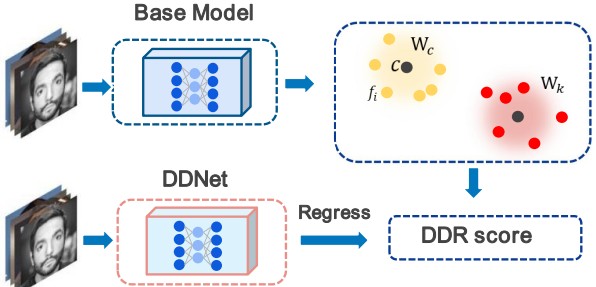

Figure 1: The pipeline of group representation learning with our DDL. Given a base feature extract model, we first compute the DDR score for each training element and then train a light-weight discriminability distillation network (DDNet) to distill it. During testing, the DDNet will predict discriminability for each element in the test set.

$s$ denotes the size of $\mathcal{X}$, the intra-class distance and inter-class distance are formulated as

$$
\begin{aligned}
\mathcal{C}_{ic} &= \frac{f_i \cdot W_c}{\|f_i\|_2 \|W_c\|_2}, \\
\mathcal{C}_{ij} &= \frac{f_i \cdot W_j}{\|f_i\|_2 \|W_j\|_2}, \; j \neq c.
\end{aligned}
\tag{2}
$$

The intra-class distance $\mathcal{C}_{ic}$ and inter-class distance $\mathcal{C}_{ij}$ are shown in Figure 2. After training the base model on classification task, features of the elements from the same class are projected to hyperspace tightly in order to form an explicit decision boundary. Furthermore, elements close to the centroid of their corresponding class are the representative ones, while elements far away from their corresponding class or closer to other classes are not discriminative enough. Based on this observation, we define the *discriminability* $Q_i$ of $f_i$ as:

$$
Q_i = \frac{\mathcal{C}_{ic}}{\max \{\mathcal{C}_{ij} \mid j \in [1, C], j \neq c\}},
\tag{3}
$$

i.e., the ratio of feature distance between the centroids of its class and the hardest-negative class. Considering the variant number of elements in different groups, we further normalize the *discriminability* by:

$$
\mathcal{D}_i = \tau \left( \frac{Q_i - \mu(\{Q_j \mid j \in [1, s]\})}{\sigma(\{Q_j \mid j \in [1, s]\})} \right)
\tag{4}
$$

where $\tau(\cdot)$, $\mu(\cdot)$ and $\sigma(\cdot)$ denote the sigmoid function, the mean value and the standard deviation value of $\{Q_j \mid j \in [1, s]\}$, respectively. We denote $D_i$ as the discriminability distillation regulation (DDR) score.

Cooperated with the feature space distribution, the DDR score $\mathcal{D}_i$ is more interpretable and reasonable than the quality score in traditional quality learning. It can discriminate features better by explicitly encoding the intra- and inter-class distances with class centroids.

## 3.3 DISCRIMINABILITY DISTILLATION LEARNING

From section 3.2, given a training dataset $\theta$ and a corresponding model, the DDR score $\mathcal{D}_i$ of $f_i$ can be naturally computed by Eq (2)-(4). However, in group representation learning, $\mathcal{D}_i$ is unavailable to test set $\mathcal{T}$ without $W_j$. In order to better embed $\mathcal{D}_i$ to the group representation learning procedure, we introduce the discriminability distillation learning (DDL) to generate an approximated DDR score $\hat{\mathcal{D}}_i$ for $\mathcal{D}_i$. Given an arbitrary CNN architecture $\mathcal{N}$ and an image $I_i$, $\hat{\mathcal{D}}_i$ can be approximated by

$$
\hat{\mathcal{D}}_i = \mathcal{N}(I_i; \boldsymbol{\theta}),
\tag{5}
$$

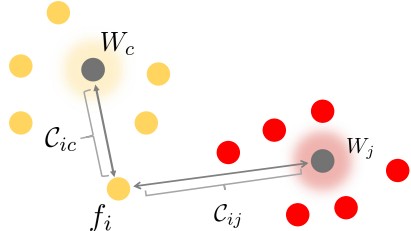

Figure 2: The formulation of discriminability distillation regulation. It is the ratio of element feature distance with its class centroids and the hardest-negative class's. After training the base model with classification task, element features from the same class are projected to hyperspace tightly in order to form a clear decision boundary. While the outliers who lie far away from its centroids or closer to other classes are of low *discriminability*.

where $\boldsymbol{\theta}$ is the parameter of the light-weight DDNet $\mathcal{N}$. At the training stage, we apply mean squared error between $\hat{\mathcal{D}}_i$ and target $\mathcal{D}_i$ as

$$L = \frac{1}{2N} \sum_i^N (\hat{\mathcal{D}}_i - \mathcal{D}_i)^2 \tag{6}$$

where $N$ is the batch size.

### 3.4 Feature Aggregation $\mathcal{G}$

At the test stage, we can generate $\hat{\mathcal{D}}_i$ via Eq (5) for each test element $I_i$ in the given element set $I_S$. The feature aggregation $\mathcal{G}$ in Eq (1) can be formulated as

$$F_{I_S} = \mathcal{G}(f_1, f_2, \cdots, f_n) = \sum_i^n \frac{\hat{\mathcal{R}}_i f_i}{\hat{\mathcal{R}}_i}, \tag{7}$$

where $n$ is the element number of $I_S$, and $\hat{\mathcal{R}}_i$ is the re-scaled DDR score of $\hat{\mathcal{D}}_i$ via

$$\hat{\mathcal{R}}_i = K\hat{\mathcal{D}}_i + B. \tag{8}$$

In Eq (8), we scale the DDR score of element set $I_S$ between 0 and 1 to ensure a same range for element sets with different lengths. $K$ and $B$ are formulated as

$$K = \frac{1}{\max\{\hat{\mathcal{D}}_i \mid i \in [1, n]\} - \min\{\hat{\mathcal{D}}_i \mid i \in [1, n]\}}, \tag{9}$$

$$B = 1 - K\max\{\hat{\mathcal{D}}_i \mid i \in [1, n]\}. \tag{10}$$

### 3.5 Advantage of Discriminability Distillation Learning

Different from the subjective quality judgment of an image or quality learning via attention mechanism, we explicitly assign *discriminability* for each element via the feature space distribution. By jointly considering the inter- and intra-class distances with class centroids, DDL can effectively approximate how discriminative a feature is. According to aggregating more information from features with high *discriminability* in an element set, the recognition performance can be significantly boosted. With the assistant of DDL, both hard samples (the model is easy to fail on them) or low-quality images (the model is easy to give wrong predict results) can be weakened. Moreover, the well design discriminability distillation learning process needn't modify the base model, making it easy to be plugged into many popular recognition frameworks.

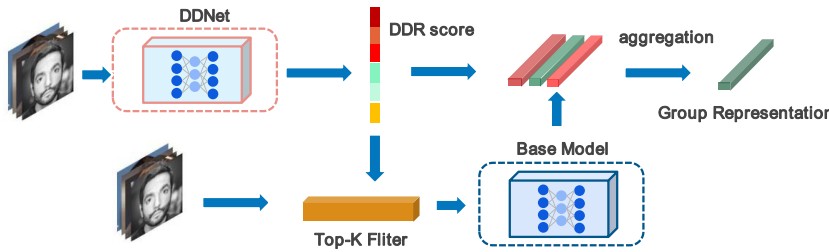

Figure 3: The pipeline of the test stage of DDL. For a set of elements, we first predict DDR score by the trained-well DDNet for each element. Then the feature extracted by the base model will be weighted aggregation to form the group representation. To achieve accuracy and computation consumption trade-off, a top-K filter can also be applied to select the most discriminative elements and only extract features and aggregate with them.

## 4 EXPERIMENTS

### 4.1 SET-TO-SET FACE RECOGNITION

#### DATASETS AND PROTOCOLS

In this section, we evaluate DDL for set-to-set face recognition on four datasets including two video sets: the YouTube Face (YTF) Wolf et al. (2011), iQIYI-VID-FACE iQIYI (2019); and two template-based sets: the IARPA Janus Benchmark A (IJB-A) Klare et al. (2015) and the IARPA Janus Benchmark C (IJB-C).

**YTF**: The YouTube Face dataset includes 3425 videos of 1595 identities with an average of 2.15 videos per identity. The videos vary from 48 frames to 6,070 frames. We report the 1:1 face verification rate of the given 5,000 video pairs in our experiments without fine-tuning.

**iQIYI-VID-FACE**: The iQIYI-VID-FACE dataset iQIYI (2019) aims to identify the person in entertainment video by face images. It is the largest video face recognition test benchmark so far, containing 643,816 video clips of 10,034 identities. The test protocol is 1:1 verification and the True Positive Rate (TAR) under False Positive Rate (FAR) at 0.01% is reported.

**IJB-A**: The IARPA Janus Benchmark A (IJB-A) Klare et al. (2015) is an unconstrained face recognition benchmark, containing 25, 813 faces images of 500 identities. It is a template-based test benchmark where both still images and video frames are included in templates. We report 1:1 verification results by following official protocol.

**IJB-C**: The IARPA Janus Benchmark C (IJB-C) is an extended version of IJB-A. It has 140, 740 faces images of 3, 531 subjects, so we can test under more strict False Accept Rates (FAR). Since the images in IJB-C dataset have large variations, it is regarded as a challenging set-to-set face recognition benchmark.

#### THE DETAILS OF DISCRIMINABILITY DISTILLATION NETWORK

All set images will be sent to the DDN at the inference stage, so the DDN is sensitive to computational burden. To make the inference stage efficient, the DDN is designed to be light-weighted. The typical DDN is a channel reduced ResNet-18 network, which only introduces 81.9 Mflops computation for $112 \times 112$ input. Compared with base feature extraction models which often reach 24 Gflops (ResNet-101), it is super-efficient. More discussions on the DDN architecture can be found in the Appendix A.

#### EVALUATION ON YOUTUBE FACE AND IQIYI-VID-FACE

As shown in Table 1, our DDL achieves state-of-the-art performance on the Youtube Face benchmark. It outperforms Deng et al. (2019a) by 1.17% and other set-to-set face recognition methods by impressive margins. Since we train a stronger base model, the baseline average method is higher

| Method | Accuracy(%) | Method | Accuracy(%) |
|---|---|---|---|
| Li et al. (2014) | 84.8 | Taigman et al. (2014) | 91.4 |
| Schroff et al. (2015) | 95.52 | Yang et al. (2017) | 95.72 |
| Sun et al. (2015) | 93.20 | Liu et al. (2017c) | 96.17 |
| Gong et al. (2019) | 96.50 | Rao et al. (2017b) | 96.52 |
| Liu et al. (2019b) | 96.21 | Rao et al. (2017a) | 94.28 |
| Wang et al. (2018b) | 97.6 | Deng et al. (2019a) | 98.02 |
| *Average* | 98.27 | *Top 1* | 97.11 |
| | | DDL | **99.19** |

Table 1: Video face verification performance on YouTube Face dataset, compared with state-of-the art methods and baseline methods.

| Method | TPR@FPR=1e-4(%) | Method | TPR@FPR=1e-4(%) |
|---|---|---|---|
| MSRA | 71.59 | Alibaba-VAG | 71.10 |
| Insightface | 67.00 | DDL (PolyNet) | **72.98** |
| *Average* | 65.84 | *Top 1* | 65.22 |
| | | DDL | **69.05** |

Table 2: Comparsion with different participants and aggreation stragey on the IQIYI-VID-FACE challenge. By combing with PolyNet, DDL achieves state-of-the-arts peformance. (http://www.insightface-challenge.com/results)

than Deng et al. (2019a), but for comparison with different aggregation strategies like average pooling, DDL can also boost performance by 0.92%, which indicates DDL has learned a meaningful pattern for discriminability. As a post-training module, DDL can cooperate with any pre-trained model. To prove the robustness and good generalization ability of our DDL, we re-implement more baseline and combine them with DDL, the results are shown in the Appendix A.

Since the results on YouTube Face benchmark tend to be saturated, we test our DDL on the challenging video face verification dataset IQIYI-VID-FACE, which is of large scale and more close to industrial application. Compared with the average pooling, DDL improves performance by 3.21%. Even aggregating the top 1 DDR score features only can still achieve the performance of average aggregation for all frames. It shows that our DDL has selected the most discriminative element of the set. By combining PolyNet base model, our DDL achieves the state-of-the-arts performance on the IQIYI-VID-FACE dataset.

What's more, the good performance of only aggregating top 1 discriminability frame shows that we needn't extract all frames by a heavy base model, but only the most discriminative one during inference. Since the computational consumption for DDNet is efficient, usually only one-tenth of other heavy models. It can be seen that DDL makes a huge difference for industrial video face recognition application.

EVALUATION ON IJB-A AND IJB-C

Tables 3 and 4 show the results on the IJB-A and IJB-C dataset for different methods. From the two tables, we can see that our DDL improves verification performance by a convincing margin with average pooling for both two benchmarks, especially under severe FAR at 0.001% by 1.94% and FAR at 0.0001% by 6.88% on IJB-C. Compared with IJB-A, IJB-C has more images and covers more variations among images, such as pose, blur, resolution, and conditions. So the performance gain with DDL is huger.

Compared with the state-of-the-art methods, our DDL improves IJB-A by 4.83% when FAR =0.001 and IJB-C by 4.89% when FAR = 0.0001%. These results indicate the effectiveness and robustness of our DDL. What's more, unlike many previous methods that need fine-tune with the base model on set-to-set recognition training datasets, the only supervision for DDL training is the feature dis-

| Method | IJB-A (TAR@FAR) | | |
|---|---|---|---|
| | FAR=0.001 | FAR=0.01 | FAR=0.1 |
| Crosswhite et al. (2018) | $83.6 \pm 2.7$ | $93.9 \pm 1.3$ | $97.9 \pm 0.4$ |
| Sankaranarayanan et al. (2016) | $81.30 \pm 2.0$ | $91.0 \pm 1.0$ | $96.4 \pm 0.5$ |
| Xie & Zisserman (2018) | $92.0 \pm 1.3$ | $96.2 \pm 0.5$ | $98.9 \pm 0.2$ |
| Liu et al. (2017c) | $89.31 \pm 3.92$ | $94.2 \pm 1.53$ | $98.02 \pm 0.55$ |
| Cao et al. (2018) | $92.1 \pm 1.4$ | $96.8 \pm 0.6$ | $99 \pm 0.2$ |
| Yang et al. (2017) | $88.1 \pm 1.1$ | $94.1 \pm 0.8$ | $97.8 \pm 0.3$ |
| Liu et al. (2017c) | $89.31 \pm 3.92$ | $94.2 \pm 1.53$ | $98.02 \pm 0.55$ |
| Zhong et al. (2018) | $93.5 \pm 1.5$ | $97.2 \pm 0.3$ | $99.0 \pm 0.2$ |
| Liu et al. (2019b) | $93.61 \pm 1.51$ | $97.28 \pm 0.28$ | $98.94 \pm 0.31$ |
| Deng et al. (2019a) | $97.89 \pm 1.5$ | $98.51 \pm 0.3$ | $99.05 \pm 0.2$ |
| *Average* | $97.71 \pm 0.6$ | $98.43 \pm 0.4$ | $99.01 \pm 0.2$ |
| DDL | $\mathbf{98.44 \pm 0.3}$ | $\mathbf{98.79 \pm 0.2}$ | $\mathbf{99.13 \pm 0.1}$ |

Table 3: Peformance comparsions on IJB-A verificatiton benchmark. The True Accept Rates (TAR) vs. False Postive Rate (FAR) are reported.

| Method | IJB-C (TAR@FAR) | | | | |
|---|---|---|---|---|---|
| | 0.0001% | 0.001% | 0.01% | 1% | 1% |
| Yin et al. (2019) | - | - | - | 0.1 | 83.8 |
| Xie et al. (2018) | - | - | 88.5 | 94.7 | 98.3 |
| Zhao et al. (2019) | - | 82.6 | 89.5 | 93.5 | 96.2 |
| Xie & Zisserman (2018) | - | 77.1 | 86.2 | 92.7 | 96.8 |
| Cao et al. (2018) | - | 74.7 | 84.0 | 91.0 | 96.0 |
| Shi & Jain (2019) | - | 89.64 | 93.25 | 95.49 | 97.17 |
| Deng et al. (2019a) | 86.25 | 93.15 | 95.65 | 97.20 | 98.18 |
| *Average* | 84.06 | 93.81 | 95.73 | 96.98 | 97.87 |
| DDL | **91.14** | **95.75** | **96.94** | **97.72** | **98.36** |

Table 4: Peformance comparsions on IJB-C verificatiton benchmark. The True Accept Rates (TAR) vs. False Postive Rate (FAR) are reported.

criminability generated with the base model on the same training set. So the fine-tune process is not needed.

To qualitatively evaluate the discriminability pattern learned by our DDL, we visualize the discriminability score distribution for two template images in IJB-C datasets. As shown in Figure 4, DDL can effectively identify image discriminability. Pictures with large poses, visual blur, occlusion, and incomplete content are regarded to be low discriminative. The efficient discriminability judgment ability for our DDL leads to an extraordinary performance on set-to-set face recognition problem. More visualization examples can be found in the Appendix A.1.

## 4.2 VIDEO ACTION RECOGNITION

### DATASETS

**ActivityNet-1.2.** The ActivityNet-1.2 dataset Caba Heilbron et al. (2015) contains 4,819 training videos and 2,383 validation videos for 100 action class. The duration of those videos varies from 2 seconds to 4 minutes and the average length of videos is 2 minutes. We carefully remove videos with more than one label. Frames are extracted from videos and their width is resized to 240 pixels. It is an untrimmed dataset, so large visual and content variation exists.

**Kinetics-700.** Kinetics Kay et al. (2017) is a well-trimmed and popular action recognition benchmark. Since around 2% videos of Kinetics-600 and 12% of videos of Kinetics-400 are not accessi-

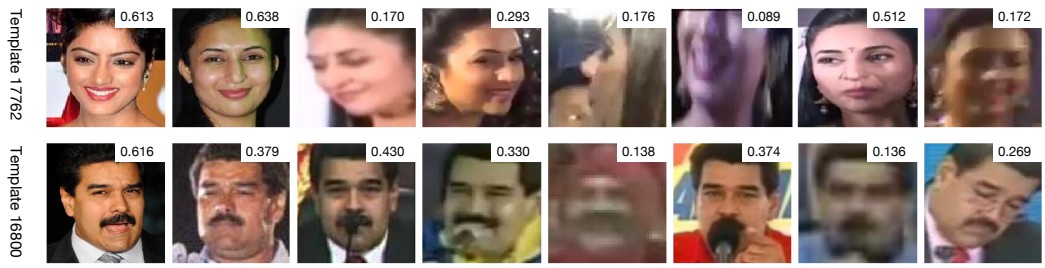

Figure 4: The visualization results of discriminability for images of Template ID 17762 and 16800 from IJB-C dataset.

ble from youtube, we conduct experiments on the Kinetics-700 dataset. Kinetics-700 contains 700 classes and 650k videos, and each action class has at least 600 video clips. Each clip is human annotated with a single action class and lasts around 10s.

BASE MODEL AND TEST PROTOCOL

To combine and compare with DDL, we choose three clip-based 3D CNN models 3D-ResNet-50 Hara et al. (2018), SlowFast-50 Feichtenhofer et al. (2018) and R(2+1)D-50 Tran et al. (2018). For testing, they firstly split the whole videos into clips and inference them individually. Then clip-level results will be average pooled to get the whole video's prediction. All three methods hugely rely on the dense sampling strategy during testing, which is quite computational consumed.

For our DDL, we firstly filter clips by the light-weight DDNet. Then we only extract and aggregate features from the top-K discriminability clips. The K value selected for Activitiynet-1.2 and Kinetics-700 is 9 and 5 respectively, for that they achieve the best efficiency and accuracy trade-off in our experiments. For comparison, we randomly or uniformly sample K clips in videos. Dense sampling is also used to compare.

EVALUATION ON ACTIVITYNET-1.2

As shown in Table 5, DDL improves recognition performance for all baseline models. For the state-of-the-art clips-based model SlowFast, combining it with DDL can achieve around 4% accuracy gain compared with randomly or uniformly sample. What's more, DDL can even outperform dense evaluation along with the whole video by a huge margin. It is reasonable as DDL can select and aggregate clips with the highest discriminability, which avoids video clips with visual blur or ambiguous content to pollute the video-level prediction.

What's more, combining with DDL, the inference process can be hugely accelerated. For testing a two minute videos with SlowFast-50 model as example, original dense sampling strategy introduces 17 Glops $\times$ 60 computation , but for DDL enhanced, only top discriminability videos need to be predicted by the heavy base model. The total consumption is 1.7 Glops$\times$ 60 for assessing the discriminability plus 9$\times$17 Gflops for extracting the top discriminability clip feature, which achieves 4x speed up.

EVALUATION ON KINETICS700.

As shown in Table 6, DDL outperforms random sampling by 1.84% and uniform sampling by 2.18%. For dense sampling, DDL can achieve 0.86% gain with 6x speed up. Since Kinetics-700 is well trimmed, the improvement is not so significant compared with the untrimmed video dataset ActivityNet-1.2. However, as a post-training module, DDL can be incorporated with any clip-based 3D action recognition model without base model re-trained. It can achieve more efficient and accurate inference, which is very suitable for real-world video understanding applications.

| Model | DDL (9 clips) | | Random ( 9 clips) | | Uniform (9 clips) | | Dense ( all clips) | |
|---|---|---|---|---|---|---|---|---|
| | Acc(%) | GFLOPs×clips | Acc(%) | GFLOPs×clips | Acc(%) | GFLOPs×clips | Acc(%) | GFLOPs×clips |
| 3D-RS-50 | 86.38 | 37×9+1.7×60 | 82.83 | 37×9 | 83.14 | 37×9 | 83.92 | 37×60 |
| R(2+1)D-RS-50 | 89.08 | 39×9+1.7×60 | 84.51 | 39×9 | 84.89 | 39×9 | 85.46 | 39×60 |
| SlowFast-RS-50 | **90.21** | 16×9+1.7×60 | 85.92 | 16×9 | 86.14 | 16×9 | 87.72 | 16×60 |

Table 5: Video action recognition results on ActivityNet-1.2 dataset. We compare randomly, uniformly and filter by DDL to select 9 clips to aggregate. Accuracy is reported on the valiation set. And the compute consumption is estimated on a 2 minutes videos (above 60 clips), for that 2 minutes is the average video length of ActivityNet-1.2.

| Model | DDL (5 clips) | | Random ( 5 clips) | | Uniform (5 clips) | | Dense ( 30 clips) | |
|---|---|---|---|---|---|---|---|---|
| | Acc(%) | GFLOPs×clips | Acc(%) | GFLOPs×clips | Acc(%) | GFLOPs×clips | Acc(%) | GFLOPs×clips |
| 3D-RS-50 | 71.01 | 37×5+1.7×30 | 68.26 | 37×5 | 67.43 | 37×5 | 68.83 | 37×30 |
| R(2+1)D-RS-50 | 72.51 | 35×5+1.7×30 | 69.24 | 35×5 | 68.79 | 35×5 | 70.94 | 35×30 |
| SlowFast-RS-50 | **74.23** | 16×5+1.7×30 | 72.39 | 16×5 | 72.05 | 16×5 | 73.37 | 16×30 |

Table 6: Video action recognition results on Kinetics-700 dataset. We compare randomly, uniformly and filter by DDL to select 5 clips to aggregate. Accuracy is reported on the validation set and is the average of top1 and top 5 accuracy. Dense sample strategy sample 10 clips along the temporal axis and random crop 3 clips on the spatial axis.

CROSS DATASET AND MODEL EVALUATION.

In the last two sections, we train specialized DDNet for different datasets and base models, respectively. To demonstrate the generalization ability of our DDL, we conduct the cross dataset and cross model evolution experiment. Action recognition model R(2+1)D for ActivityNet-1.2 dataset is combined with DDL trained with Kinetics-700 and base model SlowFast, the results are shown in Table 7. It can be found that through training with the different dataset and base model causes a performance drop, DDL can still improve about 1% accuracy for dense sampling. These results show that the discriminability pattern learned by DDL is robust and well to generalize to other datasets. What's more, the generalization ability can avoid training the DDL model for each action model and each dataset, making the DDL more efficient and practical.

| Base Model, Datatset | Accuracy(%) | Base Model, Datatset | Accuracy(%) |
|---|---|---|---|
| SlowFast-RS-50, ActivityNet-1.2 | 87.91 | R(2+1)D, ActivityNet-1.2 | 89.08 |
| SlowFast-RS-50, Kinetics-700 | 86.31 | R(2+1)D, Kinetics-700 | 86.92 |
| Uniform | 84.51 | Dense | 85.46 |
| Random | 84.80 | | |

Table 7: Results for cross dataset and cross model experiment. All results are reported with base action recognition R(2+1)D-ResNet-50 on ActivityNet-1.2 dataset. The training procedures for DDL vary from dataset and base model.

# 5 CONCLUSION

In this paper, we have proposed a novel post-processing module called DDL for all group-based recognition tasks. We explicitly define the discriminability with observations on feature embedding, then apply a light-weight network for discriminability distillation and feature aggregation. We identify the advantage of our proposed methods in the following aspects: (1) The entire discriminability distillation is performed without modifying the pre-trained base network, which is highly flexible comparing with existing quality-aware or attention methods. (2) Our distillation network is extremely light-weighted with great computational cost salvage. (3) With our DDL and feature aggregation, we achieve state-of-the-art results on set-to-set face recognition and action recognition tasks.

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

## A  APPENDIX

IMPLEMENTATION DETAILS FOR SET-TO-SET FACE RECOGNITION

**Pre-processing**: RetinaFace Deng et al. (2019b) is used to detect faces and their corresponding landmarks for all datasets. Images are aligned to $112 \times 112$ by similarity transformation based on their landmarks.

**Training**: We train our base recognition model and DDL on MS-Celeb-1M dataset Guo et al. (2016). Since the original dataset has been proved to be dirty, we use the cleaned version by Deng et al. (2019a), which consists of 5,179,510 pictures from 93,431 identities. For the base model training, we select the modified version of ResNet-101 He et al. (2016) introduced by Deng et al. (2019a) and PolyNet Zhang et al. (2017b) as the backbone network.The loss function is ArcFaceDeng et al. (2019a). The learning rate starts from 0.1 and is divided by 10 after 60k, 80k iterations, and we finish our training at 11k. The base model is trained on 16 GPUs with a total batch size of 1024 and based on Pytorch Paszke et al. (2017) framework. After training the base network, we extract features for each picture from the training set and compute its DDR score as discussed in section 3.2. Then the light-weight DDNet will regress the score by L2 loss with the same training image input. The learning rate for training our DDNet starts from 0.1 and is divided by 10 after 30k, 60k, and 80k iterations, and the total number of iterations is 100k. Linear warm-up with 10k iterations is applied towards stable training. The DDNet is trained on 8 GPUs with a total batch size of 512 are used.

DISCUSSION OF DDNET ARCHITECTURE

As has discussed in section 4.1, the design principle of our DDNet is computation efficient, to introduce less computation burden. The typical architecture of DDNet is shown in Table 8, which is a channel reduction Resnet-18 network. We also conduct experiments on deeper DDNet which has 34 layers and wider DDNet which is a standard Resnet-18 network. As shown in Table 9, neither the deeper or the wider network leads to better performance, which illustrates that the discriminability of face images can be well learned with limited parameters. As for action recognition, we utilize the same DDNet architecture where the 2D convolution kernel is replaced by 3D.

| stage | DDNet | output sizes $H \times W$ |
|---|---|---|
| raw input | | 112×112 |
| $conv_1$ | $3 \times 3, 8$ 
 stride $1, 1$ | 112×112 |
| $res_2$ | $\begin{bmatrix} 1 \times 1, 8, \text{stride2}, 2 \\ 1 \times 1, 8, \text{stride1}, 1 \end{bmatrix} \times 2$ | 56×56 |
| $res_3$ | $\begin{bmatrix} 1 \times 1, 16, \text{stride2}, 2 \\ 1 \times 1, 16, \text{stride1}, 1 \end{bmatrix} \times 2$ | 28×28 |
| $res_4$ | $\begin{bmatrix} 1 \times 1, 32, \text{stride2}, 2 \\ 1 \times 1, 32, \text{stride1}, 1 \end{bmatrix} \times 2$ | 14×14 |
| $res_5$ | $\begin{bmatrix} 1 \times 1, 48, \text{stride2}, 2 \\ 1 \times 1, 48, \text{stride1}, 1 \end{bmatrix} \times 2$ | 7×7 |
| global average pool, fc | | DDR score |

Table 8: The architecture of our light-weight DDNet. Typical DDNet is a channel reduction Resnet-18 network and only introduces 81.9 Mflops computation, which is super efficient.

| Architecture | IJB-C(TAR@FAR) | | | | |
|---|---|---|---|---|---|
| | 0.0001% | 0.001% | 0.01% | 1% | 1% |
| ResNet-18 | 90.93 | 95.74 | 96.92 | **97.73** | 98.35 |
| ResNet-34-CD | 91.13 | 95.74 | 96.90 | 97.72 | 98.33 |
| ResNet-18-CD | **91.14** | **95.75** | **96.94** | 97.72 | **98.36** |

Table 9: Peformance comparsions for different architecture on IJB-C verificatiton benchmark. The True Accept Rates (TAR) vs. False Postive Rate (FAR) are reported. CD represents channel reduction as shown in Table 8.

RESULT OF COMBINING MORE BASELINE WITH DDL ON YOUTUBEFACE

The baseline model we select in the main part is trained with ArcFace loss function Deng et al. (2019a). We also re-implement more baseline model with different loss function like Wang et al. (2018b), Liu et al. (2017a). As shown in Table 10, when enhanced by DDL, all baseline achieve performance above 1% performance gain, which indicates the good robustness and generalization ability of our DDL.

DISCUSSION ON UNBALANCED DATA CLASS WHEN TRAINING DDL

Since the data imbalance between classes and such a long-tail distribution has become concerns for training recognition models Zhang et al. (2017a); Van Horn & Perona (2017), we conduct experiments with two settings for MS1M dataset Guo et al. (2016) :

- Tail data: We use one-tenth of raw data with the minimum sample count in each class. It has insufficient data per class and contains 519,379 pics from 29,210 classes.
- Head data: We use one-tenth of raw data with the maximum sample count in each class. It has enough data per class and contains 544,381 pics from 4,833 classes.

| SF | SF(DDL) | CF | CF(DDL) | AF | AF(DDL) |
|------|---------|------|---------|------|---------|
| 95.8 | 97.1 | 97.8 | 98.7 | 98.3 | 99.2 |

Table 10: Combine DDL with more baselines including SphereFace (SF) and CosFace (CF). All baseline models are trained with same backbone and datasets. Average pooling is used as the default aggregation strategy.

The comparison results for the two settings on IJB-C datasets are illustrated in Table 11. The performance for tail data and head data are almost the same because the formulation of DDR doesn't explicitly encodes identity information. For the classification task, insufficient data for one class may lead to bad representation for that class in subspace, but for our DDL training, the DDR definition is class-unspecified, which means only the feature's intra- class and inter-class distance matters, but not the identity information.

What's more, the slight performance drop for training with one of the tens of amount indicates that by increasing the training sample, the performance of DDL can further improve. But around 500,000 data are enough for training a good DDL model.

| Data | IJB-C(TAR@FAR) | | | | |
|------|---------|--------|-------|-------|-------|
|      | 0.0001% | 0.001% | 0.01% | 1% | 1% |
| Tail | 89.81 | 95.38 | 96.62 | 97.48 | 98.14 |
| Head | 89.92 | 95.41 | 96.59 | 97.51 | 98.13 |
| Full | **91.14** | **95.75** | **96.94** | **97.72** | **98.36** |

Table 11: Peformance comparsions for discussion of class data imbalance when training DDL. Head and tail data setting are of near same number samples, but varies from class numbers.

RESULTS OF TRAINING DDL WITH OTHER DATASETS.

In the main article, we train our base model and DDL on the MS1M dataset Guo et al. (2016), which is the largest open face datasets. In this section, we also train our base model and DDL on the IMDB dataset Wang et al. (2018a), which contains 59k identities and 1.7M images. The results on IJB-C dataset are in Table 12. Training with the smaller dataset, DDL also achieves competitive results and better than average pooling for a huge margin. At FPR=0.01%, compared with average pooling, DDL boosts the performance by 29.64%. This indicates that a weak model can benefit more from DDL.

| Method | IJB-C(TAR@FAR) | | | | |
|--------|---------|--------|-------|-------|-------|
|        | 0.0001% | 0.001% | 0.01% | 1% | 1% |
| *average* | 0.002 | 0.22 | 39.39 | 91.55 | 96.34 |
| DDL | 0.002 | **0.40** | **66.33** | **94.83** | **97.55** |

Table 12: Peformance for training both base model and DDL on IMDB dataset on IJB-C benchmark. The True Accept Rates (TAR) vs. False Postive Rate (FAR) are reported.

RESULTS OF CASCADE TRAINING WITH DDL

Applying cascade training to refine a better centroid could be a reasonable idea. We conduct experiments. After training one DDL, we use the DDR scores to weighted refine each class's centroid and generate new DDR scores for DDNet to regress. The results are in Table 13. By cascade training and refined centroids, DDL boosts performance for 0.4% at FAR=0.0001%, which means good centroids leading to a better discriminability representation.

| Method | IJB-C(TAR@FAR) | | | | |
|---|---|---|---|---|---|
| | 0.0001% | 0.001% | 0.01% | 1% | 1% |
| DDL | 91.14 | 95.75 | 96.94 | 97.72 | 98.36 |
| cascade training | 91.53 | 95.91 | 97.01 | 97.82 | 98.40 |

Table 13: Peformance comparsions for cascade training on IJB-C dataset. By refining the class centroid and cascade training, DDL achieves performance gain.

### DISCUSSION OF NORMALIZATION AND SCORE SCALE FOR DDL

In Eq(4), we normalize the discriminability by the mean and standard deviation to deal with a variant number of elements in different groups. In our experiments, if the normalization step is removed, the DDNet can't converge.

As for linear map the minimum and maximum DDR score of the element set to 0 and 1 in Eq(8)(9)(10), this step leading to 0.4% performance gain for YTF and 1.6% performance gain for IQIYI-VID-FACE. But the scale process doesn't contribute to template-based set-to-set recognition like IJB-A and IJB-C dataset. The reason is that in a video, frames are captured in the almost same condition, so the variation of frames tends to be limited. The mapping process will separate the score distribution, making it more focuses on frames with higher discriminability. While for template-based set-to-set recognition like IJB-A and IJB-C, the variation of elements is large and the discriminability score distribution has been separated well, so there is no obvious gain when applying score re-scale. More aggregation strategy scripts can found in our GitHub.

### IMPLEMENTATION DETAILS FOR ACTION RECONGNTION

To train on ActivityNet-1.2 dataset, we pre-trained our model on Kinetics-700 datasets. Video clips with adjacent 64 frames are randomly sampled from raw videos for training and testing. For the spatial domain, we randomly crop $112 \times 112$ pixels with a shorted side randomly sampled in [128, 160] pixels. 32 NVIDIA V100 GPUS with synchronized SGD training and global BN are used to train and we find that result for typical training in one 8 GPU machine is the same. Similar to the 2D-DDL, we generate the pseudo discriminability label for each clip and use a channel reduction 3D resnet-18 network to regress the score after training the base model. The training setting for Kinetics is the same with ActivityNet-1.2, except we sample clips with adjacent 32 frames.

### A.1 QUALITATIVE ANALYSIS

In the main part, we visualize two templates from IJB-C datasets. In this subsection, we visualize more examples from YouTube Face dataset. As we can see in these experiments, our DDL has learned meaningful discriminative patterns.

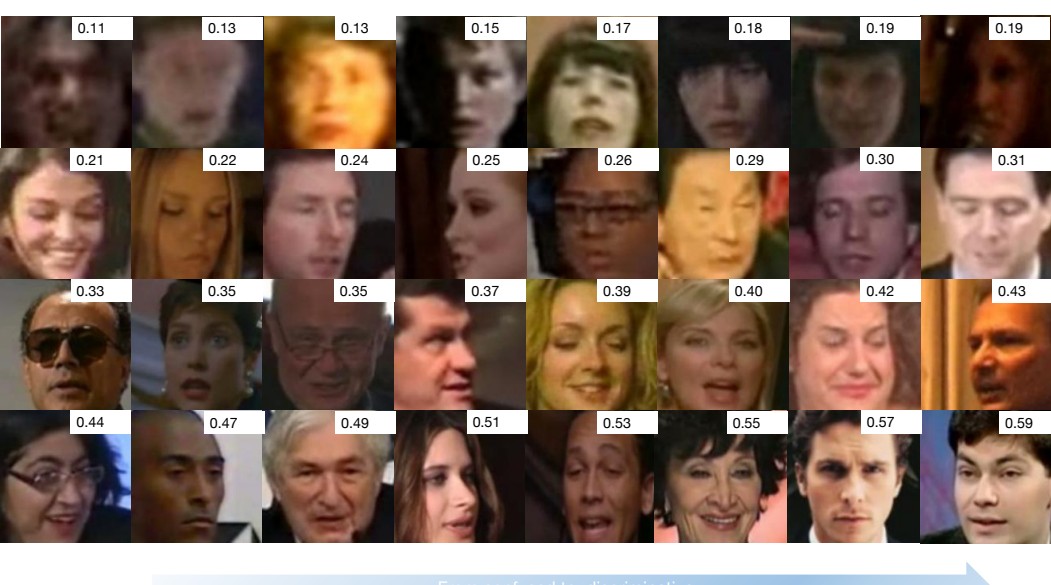

Figure 5: Visualization discrimiability of frames from different videos.

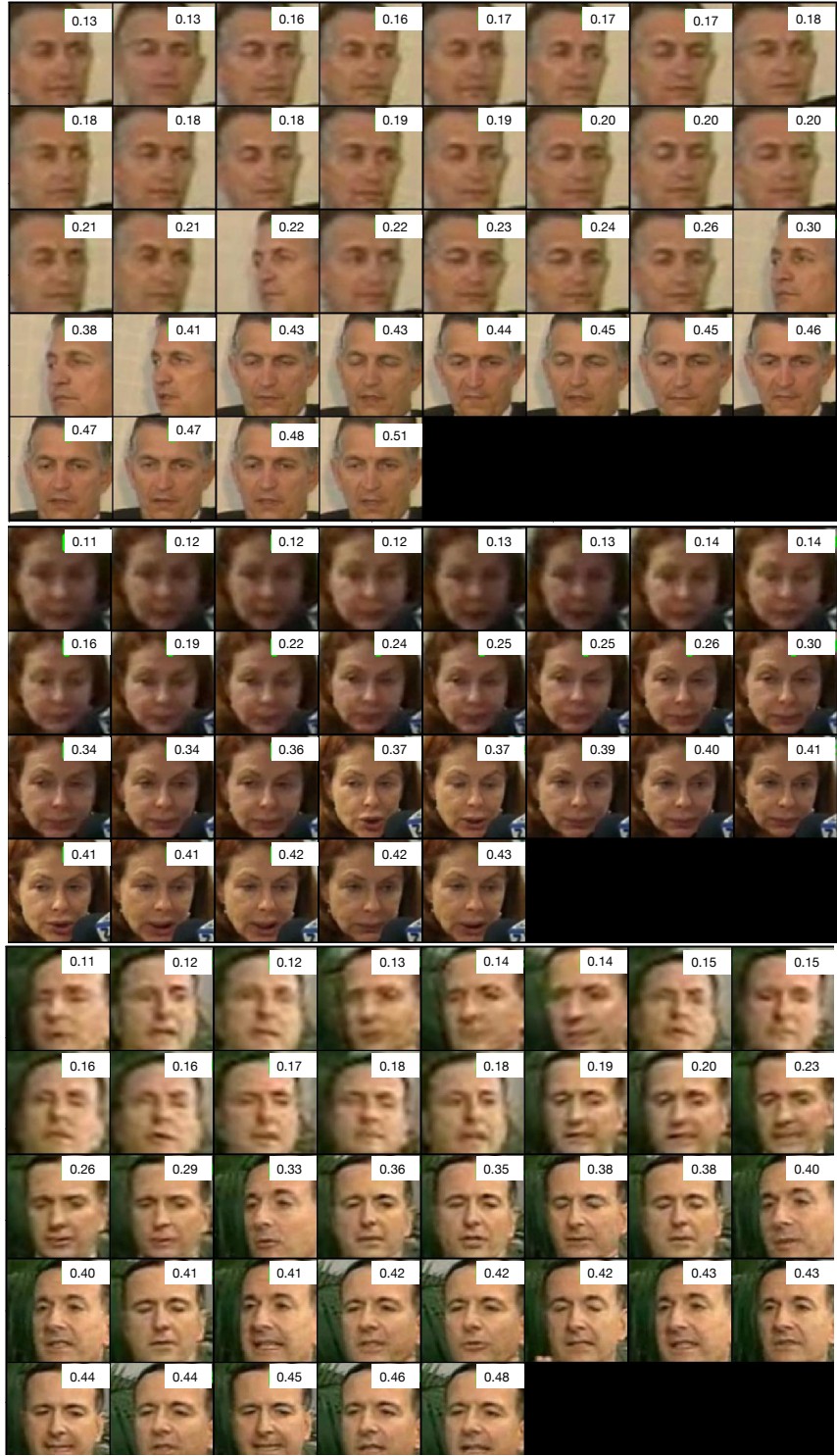

Figure 6: Visualization discrimiability of frames from one video.

