# OpenReview forum: "Discriminability Distillation in Group Representation Learning"
_ICLR.cc/2020/Conference — Reject_

### Official Review · AnonReviewer2 · 2019-10-22
**Official Blind Review #2**

**Rating:** 3

**Review:**

This paper proposes a discriminability indicator using embedded class centroids on a proxy set, and show the discriminability distribution w.r.t. the element space distilled by a light-weight auxiliary distillation network, which is called discriminability distillation learning (DDL).
This methodology was tested on a selection of datasets addressing set-to-set face recognition and action recognition. I have the following concerns, which made me to suggest rejecting this paper.

1) The methodology explained in Section 3 should be improved. In the way it is presented, it is not clear and reproducible. In detail,
1.1)  It is better to describe the pipeline together with Figure 1.
1.2) Contribution of the top part of Figure 1 in proposed pipeline is not clear. More descriptive caption should be added.
1.3) In section 1, where you ask reader to observe the feature embedding of elements which lie close to the centroid… Which section or figure are you referring to?
1.4) In subsection 2.3, clearly discuss how proposed approach is different than traditional distillation methods. Also, explain your modifications and discuss why new structure is better than default ResNet-18.
1.5) Subsections 3.2-3.4 do not contain enough explanation to support equation of proposed method.
1.6) It is not clear why authors use cosine distance, because it is simple?
1.7) The quality scores generated by DDL is very hard to interpret, which to me the one of the biggest problem of the proposed methodology.

2) The related work section should include studies more related to the motivation behind DDL (“….to select represent samples from the whole set efficiently for group understanding.”). For instance, attention models, saliency detection, key-frame detection methods, outlier detection methods, quality aware networks, etc. should be discussed and compared with DDL. Instead, authors more focused on face recognition and action recognition tasks themselves. Therefore, many related works are missing. Especially, Section 2.3., should not be a part of related work but should be integrated either to Section 1 (Introduction) or Section 3.

3) Experimental analysis:
3.1 ) It is not possible to figure out if the proposed method is really performing better than SOA. In other words, the experimental analysis should be extended. One thing authors should consider is including more datasets, especially more challenging datasets i.e., the ones not already saturated. Some examples can be: iLIDS-VID, PRID2011, YouTube Face, IJB-A benchmark.
3.2) Is there any reason to mainly focus on face recognition and action recognition?
3.3) “For face recognition, DDL can easily improve the performance by concentrating the discriminative information and for video action recognition, DDL can further accelerate the pipeline by eliminating the frames with insufficient information.” This sounds like DDL behaves differently depends on the task, but perhaps the conclusion driven by the authors cannot be fully correct given that the number of used datasets are limited to make such a general statement. I suggest authors to re-write this sentence.
3.4) The experimental analysis should include comparisons with methodologies such as quality learning via attention mechanism, etc. to better understand the necessity and effectiveness of proposed DDL. However, only for one dataset such a comparison was performed. It is strange why for other datasets the same comparison was not done.
3.5) Experimental analyses so much focuses on to justifying the performance improvements of DDL is independent to the base model selected (e.g. mainly Table 3,  but also Table 4, 5 and 6 include). However, to me this is a supporting experiment and it is more important to show the real necessity of the proposed method by comparing it with SOTA methods.
3.6) I do not find using Youtube Face benchmark suitable as it is already saturated, i.e. many existing models perform around 97%, which does not allow to show whether DDL can significantly contribute to the task or not. From the corresponding results, it looks like DDL does not provide any significant improvement.
3.7) In Table 3, it is hard to observe the contribution of DDL in terms of the performance. The improvements are in the level of 1%. More discussion is needed about that table.
3.8) “The discriminability distillation learning is more practical to untrimmed video action recognition since there are more diversity videos chip with ambiguous content and visual blur problem”  This means that DDL should not be used in some tasks, it is not generalizable, task-specific? Please elaborate it more clearly.
3.9) Table 2 is hard to interpret, improve the discussions regarding it.
3.10) Table 1 and table 2 were not referred in the text. What does DATA refer to in Table 1, why they are different?, if they are really different i.e. it is not me who misinterpret, then is not it unfair to compare the results?


4) Other suggestions:
4.1) Support your statements with references and appropriate experimental analysis.
4.2) In Section 4, only YTF dataset is mentioned but authors also use IQIYI-VID-FACE challenge dataset.
4.4) Section 4.2. is not clear at all, and should be re-written.
4.5) Some references to the tables are wrong or some are missing, pls. check them all. E.g., in section 4.2., paragraph two, table 3 should be table 2.
4.6) There is no reference to the any figures in the text.
4.7) Paper contains several typos and grammatical mistakes. Also, most of the points were not explained well.

**Experience Assessment:**

I have read many papers in this area.

**Review Assessment: Checking Correctness Of Derivations And Theory:**

I assessed the sensibility of the derivations and theory.

**Review Assessment: Checking Correctness Of Experiments:**

I carefully checked the experiments.

**Review Assessment: Thoroughness In Paper Reading:**

I read the paper thoroughly.

---

> ### Author Response · Authors · 2019-11-15
> **Response for reviewer #2**
>
> Thank you for your comments! We have carefully modified this paper based on your suggestions.
>
> (1) About the methodology explained.
>
> We modify the methodology part and re-plot the train and inference pipeline of our DDL, which is much clear. And we add references to support the observation that feature embeddings of elements tend to converge to the centroid.
>
>  The discussion of the architecture of DDNet is in the appendix, it is a lightweight Resnet-18 with reduced channel. The design principle to make it lightweight and obtain fast inference speed.
>
> As for the relationship with distillation. We are inspired by the distillation process. The typical distillation focuses on letting a student model mimic the teacher model output of the same task to achieve better training for the flops -constrained condition.  Different from this, we design the light-weight DDNet to distill the feature discriminability measured by the feature intra-class and inter-class distance and help the base model to achieve accurate and efficient group representation.  DDL is quite different from the typical distillation.
>
> We use the cosine distance since in modern feature learning tasks like face recognition, the feature is normalized and cosine distance is widely used. Other distance metrics like European distance also works and achieves near performance.
>
> About the interpretable for our DDL, we define the discriminability by the ratio of the feature's intra-class and inter-class distance.  If an element lies far from its class centroid,  it tends to be not discriminative enough. While the discriminative ones are easy to be recognized and can be projected tightly to its class centroid. We let a lightweight network distill the discriminative score and learn meaningful discriminability patterns. During inference, the trained-well DDL could evaluate what kind of elements are discriminative enough for recognition by assessing the discriminability score for each test element.
>
> About the related work, we re-write this part and more group representation works are mentioned and discussed here.
>
>
> (2)About the experiment part.
>
> We append more comparisons with SOTA methods. And two set-to-set face recognition benchmark IJB-A and IJB-C are added. Our DDL improves the performance of these benchmarks by an impressive margin. For example, for the challenging set-to-set benchmark IJB-C, we outperform the SOTA by 4.89% at FAR=0.0001%.
>
> DDL is not limited to the specific task and all group representation tasks can cooperate with DDL. Since face recognition and action recognition draws much attention in the computer vision community and well-deployed in industrial applications these years, we main conduct experiments on the two tasks.
>
>  Comparisons with quality mechanism methods and other SOTA methods on benchmark are added in the refined PDF, where you can better compare the performance of DDL with other methods.
>
> Since the Youtube Face dataset tends to be saturated, we also conduct experiments on the large-scale video datasets IQIYI-VID-FACE, IJB-B, and IJB-C. DDL outperforms SOTA by a large margin in IJB-B and IJB-C which demonstrates the advance of the proposed DDL.
>
> About the sentence "The discriminability distillation learning is more practical to untrimmed video action recognition .....".  This sentence wants to illustrate that if the test set has more confused and visual blur parts, the performance gain achieved by our DDL can be huger.  ActivityNet1.2 is an untrimmed dataset, so there are many clips with ambiguous content, filtered them by DDL can achieve super performance.  While for trimmed well datasets like Kinetics, which has been carefully cut and labeled by humans,  most clips are related to the subject and clear to be distinguished, so the performance gain by our DDL could lower. But we still achieve 0.86\% performance gain compared to with 6x speedup on Kinetics. The performance is just not so huge as 2.49% in ActivityNet, but DDL also works well. DDL is not task-specific and has good robustness and generalization in group representation tasks. We conduct extensive experiments to prove this,  which can be found in the experiment and appendix.
>
> Since some previous methods (around 2016 and 2017) use private data to train the base model, we show the DATA. But since the MS1M dataset is released, it is a common practice for the face recognition community to train the model with it, so most methods after 2018 use the MS1M dataset and the comparison tends to be fair.
>
> (3) About the writing, we follow your nice comments and more statements and references are added, please read it again.

---

### Official Review · AnonReviewer4 · 2019-10-28
**Official Blind Review #4**

**Rating:** 1

**Review:**

This paper proposed a post-processing method for improving group-based recognition tasks. Several manually designed features based on the pretrained networks are supervised trained on a light-weighted network like the teacher-student module.

This paper should be rejected because (1) the novelty of the algorithm is limited: only using the well-known intra-class distance and inter-class distance as features. Besides, the superiority of such features should be better explained; (2) Similarly, the discriminability of testing image is too complicated, such as Eq. (9, 10). Why these params are designed in such ways? Explain them; (3) Why don’t you learn the discriminability by directly operating on the extracted pretrained features? The used two distances certainly are not the best choices compared with learning methods. You should compare the performance with your methods of training under the manually-designed features. (4) The written is poor, such as Figure 6: “table ??”; Table 2: “avgerage”.


**Experience Assessment:**

I have read many papers in this area.

**Review Assessment: Checking Correctness Of Derivations And Theory:**

I assessed the sensibility of the derivations and theory.

**Review Assessment: Checking Correctness Of Experiments:**

I assessed the sensibility of the experiments.

**Review Assessment: Thoroughness In Paper Reading:**

I read the paper at least twice and used my best judgement in assessing the paper.

---

> ### Author Response · Authors · 2019-11-15
> **Response for reviewer #4**
>
> (1)About the novelty. Although the intra-class and inter-class have been explored in many loss function papers such as center loss, cosface, arcface, coco, and regularface, they are aimed to design better targets for network training and forming more compact features.
>
> As for our DDL, given a convergent model, we distill the discriminability by the ratio of intra-class and inter-class and it can be regressed by a lightweight network. The biggest contribution of our DDL is not the intra or inter-class distance definition but the well-design learning process. We find that the discriminability distribution w.r.t. the element space can be distilled by a light-weight network. Moreover, the efficiency of our DDL has been evaluated in many benchmarks including YTF, IJB-A, IJB-C, ActivityNet, Kinetics and large-scale video face recognition challenge IQIYI-VID-FACE. It improves the SOTA on the group representation problem by an impressive margin.
>
> (2)The core of the definition of discriminability is Eq3, Other equations like normalization and re-scale are for stable training and inference.
>
> (3) Using only the features with limited dimension (often 512-dim for face recognition and 1024-dim for action recognition) can’t support DDNet to learn a meaningful discriminability pattern.
>
>
> Furthermore, DDNet can deprecate some images with lower discriminability to speed up the inference stage. If estimating the discriminability by feature representation, each image will be first forwarded into the base model and the computational cost is heavy.
> As we have discussed in the article, in some application scenario which pursues computational efficiency, we can first assess the discriminability score for each element by the lightweight DDNet and only aggregate the top-K discriminative elements. This can save much computation and achieve competitive accuracy.
>
> Compared with learning quality scores by a black box attention module, our definition of discriminability and the discriminability distillation learning process is more clear and interpretable.
>
> (4) In the revised version, we re-organize the article and carefully remove grammar errors and typos, making it more rigorous and readable. Furthermore, more experiments are conducted, including IJB-A, IJB-C and some ablation study experiments.

---

### Official Review · AnonReviewer3 · 2019-11-05
**Official Blind Review #3**

**Rating:** 6

**Review:**

This paper studies how to aggregate features from group inputs. The paper proposes  Discriminability Distillation Learning (DDL) to compute the aggregation coefficients. The method assumes that each sample has a discriminability property that is directly related to the task. The authors define this property and propose to learn such property by an auxiliary network. Such a network can be used in many models without affecting their original training procedure and is able to improve the performances on many tasks, including set-to-set face recognition and action recognition. The experimental results are comprehensive and convincing.

The idea is simple yet interesting and the results are good, so I tend to give a positive rating.

But overall, the paper is not well-written, and there are some questions:

1. The paper needs more proofreadings. For example,
    a. only frames **will** high scores will be weighted and aggregated
    b. As shown in table ??
2. It seems that the proposed method is limited to tasks that have the concept of “centroids”. Many tasks may not have well-defined centroids.
3. Discriminability is also a kind of quality in some ways, and it is used in a similar way to the previous methods. The authors are encouraged to discuss more the differences to highlight the novelty and contributions. Especially after viewing Fig. 3 and 4, their boundaries become less clear.
4. I might be wrong but I didn’t see any figures referenced in the main body. Please add references and organize them better.
5. The discriminability is used only once. I’m just wondering if the authors have considered cascade training, i.e., use the discriminability information to train better centroids, which then will define discriminability better.


**Experience Assessment:**

I have read many papers in this area.

**Review Assessment: Checking Correctness Of Derivations And Theory:**

I assessed the sensibility of the derivations and theory.

**Review Assessment: Checking Correctness Of Experiments:**

I assessed the sensibility of the experiments.

**Review Assessment: Thoroughness In Paper Reading:**

I read the paper thoroughly.

---

> ### Author Response · Authors · 2019-11-15
> **Response for reviewer  #3**
>
> Thanks for your nice comments!
> (1)  In the revised version, the grammar errors and typos are carefully modified.
>
> (2) In this article,  we mainly focus on the group recognition task. The most usual definition of centroids is the mean feature of class samples ( class center ).
>
> (3) Different from the subjective quality judgment of an image or quality learning via attention mechanism, we explicitly assign discriminability for each image via the feature space distribution.  Moreover, the previous "quality" mechanism mainly focuses on large poses or visual blur problems. While for our DDL, we define the discriminability by the ratio of the feature's intra-class and inter-class distance. It is highly related to the base model. In other words, if the recognition ability of the base model is weak, maybe some clear elements will also be assigned with low discriminability.  The discriminability of elements varies from the different base models. The core insight of our discriminability distillation learning is to let a  lightweight network learn the strengths and weaknesses of the base model.  Then the lightweight network could help the base model by guiding the base model to focus more on parts it could handle well with.
>
> (4) Thanks for your suggestions, more references are added in the revised version.
> (5) The cascade training is a good idea and we conduct some experiments in the appendix. By cascade training, we get better centroids and better discriminability, leading to 0.4%  performance gain at FAR= 1E-6  on IJB-C benchmark. Thanks for your nice insight!

---

### Official Review · AnonReviewer1 · 2019-11-05
**Official Blind Review #1**

**Rating:** 6

**Review:**

In this paper, the authors proposed a discriminability distillation learning (DDL) method for the group representation learning, such as action recognition recognition and face recognition. The main insight of DDL is to explicitly design the discrimiability using embedded class centroids on a proxy set, and show the discrimiability distribution w.r.t. the element space can be distilled by a light-weight auxiliary distillation network. The experimental results on the action recognition task and face recognition task show that the proposed method appears to be effective compared with some related methods. The detailed comments are listed as follows,

There are many grammar errors and typos in the current manuscript, such as
-	Our key insight is to explicitly design the discrimiability using embedded class centroids on a proxy set…

The authors proposed DDL based on the principle of the intra-class distance and inter-class distance. How to avoid the imbalance of the dataset, namely some classes have the insufficient data?

In Eq4, the authors adopt a normalization method. How about the influence of the model if we ignore this normalization? Some ablation study whether or not the normalization is missing.

The implementation details of the proposed framework are unclear. For example, how to fine tune the network?



**Experience Assessment:**

I have published one or two papers in this area.

**Review Assessment: Checking Correctness Of Derivations And Theory:**

I assessed the sensibility of the derivations and theory.

**Review Assessment: Checking Correctness Of Experiments:**

I assessed the sensibility of the experiments.

**Review Assessment: Thoroughness In Paper Reading:**

I read the paper at least twice and used my best judgement in assessing the paper.

---

> ### Author Response · Authors · 2019-11-15
> **Response for reviewer #1**
>
> Thanks for your comments! In the revised version, most grammar errors and typos are carefully modified. Furthermore, extra experiments are conducted on IJB-A and IJB-C to better evaluate the proposed method. As recommended,  some ablation studies are also designed.
>
> DDL is still effective for the imbalance of the dataset. To delve to this property, we modify the dataset to meet this requirement where both the classes with sufficient data and classes with insufficient data exist.
> The detailed setting can be found in the appendix. As demonstrated by the results, DDL is not affected.
>
> Normalization is important for training convergence in many tasks and if we remove this process, the training of DDL is hard to converge.  The computation of the score for each sample is independent. So it requires the normalization process to make the training stable.
>
> About the implementation details, we re-plot the training and inference pipeline in Figure 1 and Figure 2 in the revised PDF. To summarize, we first train a base model on the still image dataset, then generate the score for each image in the training datasets. A lightweight DDNet will be trained with these [image, score] pairs in the regression manner. At the inference stage, the DDNet can estimate the score for each image. Cooperated with the feature representation, the weighted sum manner is used to aggregate the features belonging to the same set/video. Different from many previous works, DDL does not require a fine-tuning stage.

---

### Decision · Program_Chairs · 2019-12-19

**Decision:**

Reject

**Comment:**

This paper proposes discriminability distillation learning (DDL) for learning group representations. The core idea is to learn a discriminability weight for each instance which are a member of a group, set or sequence. The discriminability score is learned by first training a standard supervised base model and using the features from this model, computing class-centroids on a proxy set, and computing the iter and intra-class distances. A function of these distance computations are then used as supervision for a distillation style small network (DDNet) which may predict the discriminability score (DDR score). A group representation is then created through a combination of known instances, weighted using their DDR score. The method is validated on face recognition and action recognition.

This work initially received mixed scores, with two reviewers recommending acceptance and two recommending rejection. After reading all the reviews, rebuttals, and discussions, it seems that a key point of concern is low clarity of presentation. During the rebuttal period, the authors have revised their manuscript and interacted with reviewers. One reviewer has chosen to update their recommendation to weak acceptance in response. The main unresolved issues are related to novelty and experimental evaluation. Namely, for novelty comparison and discussion against attention based approaches and other metric learning based approaches would benefit the work, though the proposed solution does present some novelty. For the experiments there was a suggestion to evaluate the model on more complex datasets where performance is not already maxed out. The authors have provided such experiments during the rebuttal period.

Despite the slight positive leanings post rebuttal, the ACs have discussed this case and determine the paper is not ready for publication.